# Serum-Soluble CD163 Levels as a Prognostic Biomarker in Patients with Diffuse Large B-Cell Lymphoma Treated with Chemoimmunotherapy

**DOI:** 10.3390/ijms25052862

**Published:** 2024-03-01

**Authors:** Aspasia Koudouna, Annita Ioanna Gkioka, Alexandros Gkiokas, Thomai M. Tryfou, Mavra Papadatou, Alexandros Alexandropoulos, Vassiliki Bartzi, Nikolitsa Kafasi, Marie-Christine Kyrtsonis

**Affiliations:** 1Hematology Section, First Department of Propaedeutic Internal Medicine, Laikon Hospital, National and Kapodistrian University of Athens’ Medical School, 11527 Athens, Greece; aspakoud@hotmail.gr (A.K.); anni.iwan.gk@gmail.com (A.I.G.); progoulantzaki@hotmail.com (A.G.); thommais@hotmail.com (T.M.T.); mavra90@yahoo.com (M.P.); al.alexandropoulos@gmail.com (A.A.); vbartzi@yahoo.com (V.B.); 2Immunology Department, Laikon Hospital, 11527 Athens, Greece; nkafassi@hotmail.com

**Keywords:** soluble CD163 (sCD163), tumor-associated macrophages (TAMs), Diffuse Large B-cell Lymphoma (DLBCL), serum samples, prognosis

## Abstract

The majority of patients with Diffuse Large B-cell Lymphoma (DLBCL) will respond to first-line treatment and be cured. However, the disease is heterogeneous, and biomarkers able to discriminate patients with suboptimal prognosis are needed. M2 CD163-positive tumor-associated macrophages (TAMs) were shown to be implicated in DLBCL disease activity regulation. Serum-soluble CD163 (sCD163) functions as a scavenger receptor for haptoglobin–hemoglobin complexes and is mostly expressed by monocytes and macrophages. Its levels are used to determine macrophage activation. We aimed to determine serum sCD163 in a sample of DLBCL patients and study eventual correlations with parameters of disease activity or survival. Serum sCD163 levels were measured in 40 frozen sera from patients diagnosed with DLBCL and 30 healthy individuals (HIs) using an enzyme-linked immunosorbent assay (ELISA). Statistical analyses were performed using SPSS version 28. The results showed that patients who achieved complete response after standard-of-care immunochemotherapy and were alive and disease-free after 12 months of follow-up but had elevated sCD163 levels (above median) at diagnosis presented a significantly worse overall survival compared to those with initial serum sCD163 levels below the median (*p* = 0.03). Consequently, serum sCD163 levels in patients with DLBCL may constitute a marker of long-term response to chemoimmunotherapy.

## 1. Introduction

Diffuse Large B-cell Lymphoma (DLBCL) is the most common subtype of non-Hodgkin lymphoma, representing about 30–40% of cases [1,2,3]. This aggressive form of lymphoma exhibits significant heterogeneity in its pathogenesis. Approximately 60% of patients respond positively to first-line treatment, while 25% experience relapse [4,5]. The tumor microenvironment (TME) plays a crucial role in DLBCL pathogenesis. The TME is composed of stromal and immune system cells such as T and B lymphocytes, tumor-associated macrophages (TAMs), myeloid-derived suppressor cells (MDSCs), tumor-associated neutrophils (TANs), natural killer cells (NKs), and dendritic cells (DCs) [6,7,8]. All these cells interact with each other in complex ways to form the TME, which supports the growth and survival of neoplastic cells [7,9]. M2 TAMs have been implicated in the pathogenesis of B lymphoproliferative diseases such as DLBCL. They promote tumor cell proliferation, invasion, metastasis, angiogenesis, ECM remodeling, immunosuppression, and the inhibition of phagocytosis [10,11]. Activated M2 TAMs express CD163 [12]. CD163 is a scavenger receptor and member of the cysteine-rich family [13,14,15]. It serves as a monocyte/macrophage-specific membrane marker and is particularly recognized as a marker for alternatively activated or anti-inflammatory macrophages [16,17,18,19].

The current landscape of DLBCL treatment and prognosis faces significant challenges, primarily owing to the absence of robust prognostic indicators and biomarkers that can guide therapeutic strategies. The integration of anti-CD20 rituximab (R) into the CHOP regimen and anti-CD79b polatuzumab (Pola) has marked a significant advancement in DLBCL therapy. However, the persistence of treatment-resistant and relapsed cases emphasizes the need for further research [20,21,22,23,24,25], and TME is a key factor in DLBCL progression, while interaction between DLBCL cells and various TME components, including macrophages, is closely linked to drug resistance [26,27,28,29].

Given the complex interplay within the TME and its influence on DLBCL progression and drug resistance, understanding the role of M2 CD163-positive TAMs could lead to the identification of novel biomarkers and therapeutic targets, potentially transforming DLBCL management and improving patient outcomes. We therefore investigated the potential relationship between serum-soluble CD163 (sCD163) and parameters of disease activity or prognosis in a DLBCL series.

## 2. Results

### 2.1. General Results

The 40 patients studied included 19 males and 21 females, with a mean age of 61.74 years (range, 18–93 years). The patients’ characteristics are shown in Table 1, and their laboratory tests at presentation are shown in Table 2.

According to the laboratory results obtained at the time of diagnosis, the median levels of hemoglobin, white blood cells, lymphocytes, and monocytes were found to be 11.68 gr/dL, 7.14 K/μL, 1.53 K/μL, and 0.57 K/μL, respectively. Notably, 67.5% of the patients exhibited monocytosis, while 25% had lymphopenia, with major lymphopenia (absolute lymphocytes below 0.5 K/μL) present in 10% of the patients. The median LDH level was 476 U/L, with 65% of patients exhibiting abnormal levels. Furthermore, 7.5% of the patients had LDH levels above 1000 U/L. Hypercalcemia and hypogammaglobulinemia were present in 7.5% and 10% of the patients, respectively. All patients with hypercalcemia and hypogammaglobulinemia had stage III or IV disease and had poor R-IPI scores.

Three patients were diagnosed with Ann Arbor stage I, four with stage II, nine with stage III, and twenty-four with stage IV. Twenty-three patients had a poor R-IPI prognostic score, whereas seventeen had a good score; however, none of the patients had very good R-IPI scores.

Upon analyzing the data for immunophenotyping, immunohistochemistry, and Han’s algorithm, it was shown that 20 patients exhibited positive staining for bcl6 (cut off 30%) and bcl2 (cut off 50%). Seven patients displayed cmyc staining with a cut off of 40%, while three patients showed a low expression of cmyc (0–39%). Additionally, MUM1/IRF4 and PAX5-positive staining was observed in 12 and 11 patients, respectively. Clonal B cells expressed CD10 in ten patients. In four patients, CD30 was positive, whereas one was positive for EBER. According to Han’s algorithm, the cell of origin was classified as germinal center (GC) in 45% of the patients and non-germinal center (activated B cell [ABC]) in 55% of the patients (Table 3).

The mean follow-up duration was 4 years (range: 2–170 months). Thirty patients were diagnosed with primary or secondary DLBCL.

The first-line treatment for 27 patients consisted of the R-CHOP chemotherapy regimen (rituximab, cyclophosphamide, doxorubicin, vincristine, prednisolone). Nine patients received reduced-intensity chemotherapy regimens because of their advanced age or comorbidities. Four patients were treated with double monoclonal antibodies (anti-CD20 and anti-CD79b) in combination with chemotherapy (rituximab, polatuzumab vedotin, cyclophosphamide, doxorubicin, and prednisolone). A complete response was achieved in twenty-four (60%) patients, and a partial response was achieved in five patients. In contrast, 11 (27.5%) were initially resistant and died within 12 months of diagnosis (Table 1). None of the responders died.

### 2.2. sCD163 Results

The median serum sCD163 value was 126,052 pg/mL (range: 92,056 to 164,912 pg/mL) in patients; it was statistically significantly higher than in healthy individuals (median: 26,826 pg/mL; range: 11,831–97,286 pg/mL; *p* < 0.001).

The 40 patients were divided into two groups based on the median sCD163 value of the cohort. Patients with higher than median serum sCD163 levels showed a trend for more unfavorable overall survival (*p* = 0.18, HR 1.4, 95% CI 16.7–43.2 for patients with values below median and 61.1–160.8 for patients with values above median). Specifically, the median overall survival for patients with high and low sCD163 levels was 19.8 months (1.6 years) and 46.23 months (3.8 years), respectively.

A subanalysis was then performed on patients that were alive after the first year (N = 29), thus excluding from the analysis the 11 deceased patients who were primary resistant. The median overall survival of this subcohort was 4.8 years, with a 5-year survival rate of 40%. Patients with serum sCD163 below the median had a statistically significant better overall survival than patients with levels above the median (Figure 1) (*p* = 0.03, HR 4.6, 95% CI 28.4–33.5 for patients with values below median and 50–287.9 for patients with values above median).

Because of the small number of patients and events, we failed to prove any prognostic value of serum sCD163 levels regarding progression-free survival (PFS) or time to next treatment (TTT).

A possible association was observed between the levels of serum sCD163 and PAX5 immunohistochemical staining (*p* = 0.016), but the number of cases studied is too small to reach a conclusion. 

## 3. Discussion

Our study included 40 patients (19 men and 21 women) with a median age of 62 years (range, 18–93 years), of whom 4 were older than 80 years. This study aimed to evaluate the clinical impact of serum sCD163 levels in patients with DLBCL at diagnosis. Sixty percent of patients had advanced-stage disease (Ann Arbor IV) and poor R-IPI scores. Additionally, 45% had an extranodal disease, 20% exhibited bone marrow infiltration, and the majority displayed an ABC-type DLBC. This patients’ population typically displays an aggressive clinical course, necessitating additional prognostic factors to guide treatment decisions. In spite of their very bad prognosis, 60% of patients achieved a complete response while 27.5% were refractory to first-line treatment. Notably, 27 patients received R-CHOP, 4 received R-Pola-CHP, and 9 received reduced-intensity chemotherapy due to advanced age or comorbidities. Over the past twenty years, rituximab-based chemoimmunotherapy has changed the DLBCL treatment landscape, and approximately 50% to 60% of patients are cured with first-line treatment, but 10% of patients are primary refractory and 25% experience a relapse [20,22]. Indeed, our cohort is not representative of previously published data, as more than half of the patients were in an advanced stage, and consequently, 27.5% of them were refractory to therapy, a percentage significantly higher than the one reported in the literature. We administered the immunochemotherapy protocol R-Pola-CHP to four patients recently diagnosed with advanced-stage disease and poor R-IPI scores based on the phase III PO-LARIX trial [24]. Polatuzumab vedotin combined with R-CHP in the frontline treatment of DLBCL was recently approved and produced a significant improvement in 2-year PFS compared to standard-of-care R-CHOP [30,31]. None of these four patients have relapsed to date. Moreover, the molecular subgroups of DLBCL display notably distinct biological characteristics, therapy responsiveness, and overall survival, which are determined by the cell of origin and serve as independent prognostic factors, as the ABC type has a shortened 5-year PFS [32,33,34,35,36].

All patients exhibited increased levels of serum sCD163, irrespective of stage, age, performance status, B symptoms, extranodal sites, and tumor burden. This finding implies that elevated levels of soluble receptor CD163 may be a common feature among patients with DLBCL. In recent years, several studies have demonstrated the roles of TAMs and CD163 in the pathogenesis of hematopoietic malignancies including T-cell leukemia/lymphoma, acute myeloid leukemia, and classical Hodgkin lymphoma [17,37,38,39,40,41,42,43]. In most studies, high CD163 expression was correlated with an inferior prognosis. Controversial findings, as well as significant heterogeneity, have been reported regarding the contribution of CD163 expression in non-Hodgkin’s lymphoma disease activity, specifically in DLBCL, mantle cell lymphoma, follicular lymphoma, and T lymphoma [44,45]. The majority of studies have compared CD163 and CD68 immunohistochemical staining intensity in tissue biopsy. The use of blood-based metric values to assess the burden of M2 TAMs (alternatively activated macrophages) has recently been established and tested. In a recent publication by Xiaoqing Sun, Siglec-5/CD163 values constituted an independent biomarker in DLBCL [46]. In addition, Nikkarinen et al. reported the impact of serum sCD163, detected by an ELISA, in patients with mantle cell lymphoma treated with immunotherapy [47].

In our study, we showed that DLBCL patients alive after 12 months of follow-up, those with serum sCD163 levels above the median had a significantly worse overall survival than patients with serum sCD163 levels below the median (*p* = 0.03), as shown in Figure 1. The “DLBCL patients’ population alive after 12 months of follow-up” exhibiting increased serum sCD163 levels included mostly high-risk DLBCL patients achieving complete remission with first-line treatment. If confirmed, our finding would imply that serum sCD163 levels above the median can eventually discriminate, within responding high-risk DLBCL patients, those who will experience a late relapse. In such cases, sCD163 levels could guide treatment decisions; for example, patients could undergo autologous stem cell transplantation immediately after remission, or some kind of maintenance could be eventually administered. In this small series, serum sCD163 failed to initially segregate the rare patients with a really aggressive and resistant disease that succumbed early and probably have a different underlying histology. Unfortunately, as shown in Table 3, immunohistochemical DLBCL tissue characterization was incomplete in some cases, and some of the patients with very aggressive disease might have in fact had the new high-grade B-cell lymphoma entity. Because of the inclusion of these patients not segregated by serum sCD163 levels, the values of the latter above median showed only a trend of worse survival in the whole cohort.

Identifying disease-related biomarkers has emerged as a crucial aspect of research aimed at facilitating the diagnosis and prognosis of DLBCL. Biomarkers serve as critical elements that inform guidelines for risk assessment, prognostic prediction, treatment response evaluation, and disease progression monitoring. According to Camicia’s comprehensive review of novel drug targets for personalized precision medicine in relapsed/refractory Diffuse Large B-cell Lymphoma, there are three types of drug resistance that can be identified: genetic resistance, resistance to chemotherapy, and tumor microenvironment (TME) cell adhesion-mediated drug resistance [48]. Based on our results and previously published data, we can hypothesize that the excessive activation of TAMs, reflected by sCD163 secretion, contributes to disease relapse. Serum sCD163 can be easily determined by an ELISA and could constitute an easily applicable biomarker if our results are confirmed in larger series.

## 4. Materials and Methods

We retrospectively included DLBCL patients diagnosed in our unit from 2010 to 2023 who had available frozen sera kept from diagnosis before treatment and who were willing to provide informed consent for the study and had at least 1 year of follow-up after diagnosis. The DLBCL diagnoses of the included patients were based on the latest WHO classification and the ICC International Classification including DLBCL NOS; GC DLBCL; ABC DLBCL; primary large B-cell lymphomas of immune-privileged sites (CNS, testis); primary cutaneous DLBCL; leg-type, EBV-positive DLBCL; DLBCL associated with chronic inflammation; primary mediastinal LBCL; mediastinal gray zone lymphoma; ALK-pos LBCL; plasmablastic lymphoma; intravascular LBCL; T-cell/histiocyte-rich large B-cell lymphoma; HGBCL with MYC and BCL2 rearrangement; and high-grade transformation of indolent B-cell lymphomas. The analysis also focused on the immunohistochemistry of tissue biopsy at diagnosis for the presence of bcl6, bcl2, cmyc, cyclinD1, pax5, eber, and the immunophenotype of neoplastic B-lymphocytes for the expression of cd30, cd10, cd3, and cd5. Additionally, full blood counts, including hemoglobin, total white blood cell count, absolute lymphocyte count, and absolute monocyte count, as well as laboratory findings such as lactate dehydrogenase (LDH), erythrocyte sedimentation rate (ESR), B2 microglobulin, gamma globulin, serum calcium, and serum albumin values, were retrieved from patients’ medical files.

Forty serum samples from DLBCL patients were collected at the time of the diagnosis and kept frozen. To ensure a comprehensive and representative sample, patients were selected based on specific inclusion and exclusion criteria. The inclusion criteria included a confirmed diagnosis of DLBCL as per the latest WHO classification, availability of frozen sera samples from diagnosis before any treatment, and consent to participate in the study with a minimum follow-up of one year post-diagnosis. The exclusion criteria encompassed prior treatment with corticosteroids, immunosuppressants, or chemotherapy drugs before diagnosis and any history of other malignancies or autoimmune diseases that could influence serum sCD163 levels. Serum s-CD163 measurements were performed using frozen sera samples from 40 patients at the time of diagnosis and frozen sera samples from 30 healthy individuals (HIs). Measurements were carried out using an enzyme-linked immunosorbent assay (ELISA) (Quantikine, Duo-Set R&D Systems, Minneapolis, MN, USA) according to the manufacturer’s instructions. The assay’s detection range is from 156 to 10,000 pg/mL. Specificity teats, as reported by the manufacturer (Human CD163, catalog number DY1697), showed no cross-reactivity or interference with mouse CD 163, CD6/Fc Chimera, or Tweak. Briefly, the Human CD163 Immunoassay is a solid-phase ELISA designed to measure the levels of soluble human CD163 in cell culture, serum, and plasma. This assay is an 8 h ELISA that utilizes natural human CD163. The method includes reagent preparation, plate preparation, and sample determination. The microplate was pre-coated with a monoclonal antibody specific for human CD163 and incubated overnight. Standards and samples were then pipetted into the wells of 96-well plates. After washing away the unbound substances, an enzyme-linked monoclonal antibody specific for human CD163 was added to the wells. This was followed by washing to remove any unbound antibody–enzyme reagent, and the substrate solution was then added to the wells. Color developed in proportion to the amount of CD163 bound in the initial step, and color development was stopped with the stop solution. The optical intensity of the color was measured using a microplate reader. All measurements were performed in duplicate. However, as all DLBCL patients displayed heightened levels of serum-soluble receptor CD163, exceeding the highest standard value, it was necessary to further dilute the sample with a calibrator diluent and repeat the assay. The dilution ratio used was 1:4.

Statistical analysis was performed using the SPSS v.28 software. Kaplan–Meier curves depicted survival, while the log-rank test was used to estimate the differences in outcomes between the subgroups that were studied. Statistical significance was set at *p* < 0.05. The Mann–Whitney test was used for non-parametric variables.

The study was approved by the institutional review board (IRB) of our hospital with protocol number 337 and date of approval 22 March 2019. All the participants provided written informed consent. Patient confidentiality was maintained throughout the study, and data anonymization was applied to protect personal information.

## 5. Conclusions

In this study, we found that serum-soluble CD163 may serve as a novel prognostic indicator in patients with Diffuse Large B-cell Lymphoma (DLBCL). Specifically, patients with lower levels of soluble CD163 may have a better prognosis, whereas those with higher levels may be more likely to experience resistance or recurrence. The levels of soluble CD163 reflect the burden of TAMs in the tumor microenvironment, whose roles have not been fully elucidated yet.

Further research is necessary to validate our results.

## Figures and Tables

**Figure 1 ijms-25-02862-f001:**
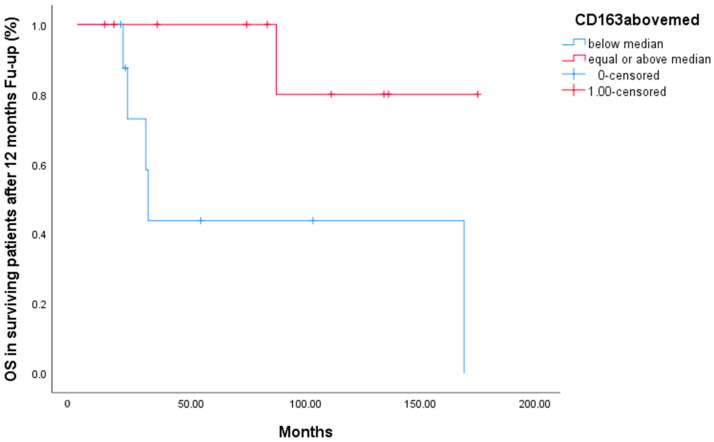
Overall survival of patients alive after 12 months follow-up.

**Table 1 ijms-25-02862-t001:** Clinical variables of all patients.

Clinical Parameters	Patients n (%)
Total	40
Median sCD163	126,052
Median Age	61.4
Gender Female Male	21 19
STAGE 1 2 3 4	3 (7.5%) 4 (10%) 9 (22.5%) 24 (60%)
R-IPI score Very good Good Poor	0 (0%) 17 (42.5%) 23 (57.5%)
Lymphadenopathy Extranodal sites Bone marrow infiltration	32 (80%) 18 (45%) 8 (20%)
Subtype GC-type Non-GC type	11 (45%) 13 (55%)
Response Complete response Partial response Refractory	24 (60%) 5 (12.5%) 11 (27.5%)
Chemotherapy protocol R-CHOP Other	27 (67.5%) 13 (32.5%)

Clinical variables of patients at the time of diagnosis. GC (germinal center).

**Table 2 ijms-25-02862-t002:** Laboratory data of all patients at the time of diagnosis.

Laboratory Data	Average Count	Total n = 40 Patient n (n %)
Hb (gr/dL) Anemia	11.68	23 (57.5%)
Absolute count WBC (K/μL)	7.14	
Absolute count Lymphocytes (K/μL) Lymphopenia (<1.2 K/μL)	1.53	10 (25%)
Absolute count Monocytes (K/μL) Monocytes > 0.5 K/μL	0.57	27 (67.5%)
LDH (U/L) UNL (>235 U/L)	476	26 (65%)
B2 Microglobulin (mg/L) UNL (>2.4 mg/L)	3.78	21 (52.5%)
Calcium serum Hypercalcemia	9.5	3 (7.5%)
Gamma Globulin Hypogammaglobulinemia	11.1	4 (10%)

Hemoglobulin (Hb), white blood cell (WBC), lactate dehydrogenase (LDH), upper normal limit (UNL).

**Table 3 ijms-25-02862-t003:** Immunophenotyping of tissue biopsy.

Immunophenotyping/Immunohistochemistry	Total n = 40 Patient n (n%)
Bcl6 (cut off: 30%) Bcl2 (cut off: 50%)	20 (50%) 20 (50%)
cmyc (cut off: 40%) Low expression (0–39%)	7 (17.5%) 3 (7.5%)
MUM1/IRF4 * (cut off: 30%) PAX5 (cut off: 5%) CD10 (cut off: 30%) CD30 CyclinD1 EBV (EBER) **	12 (30%) 11 (27.5%) 10 (25%) 4 (10%) 1 (2.5%) 1 (2.5%)
Han’s algorithm (cell of origin)	
Germinal center (GCB)	11 (27.5%)
Non-geminal center (ABC—activated B cell)	13 (32.5%)
Unknown	16 (40%)

* multiple myeloma 1/interferon regulatory factor 4 (MUM1/IRF4); ** Epstein–Barr virus (EBV)-encoded small RNAs (EBERs).

## Data Availability

Data is contained within the article.

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
