# Peer review of "Serum-Soluble CD163 Levels as a Prognostic Biomarker in Patients with Diffuse Large B-Cell Lymphoma Treated with Chemoimmunotherapy"

_ijms, 2024, doi:10.3390/ijms25052862_

Round 1

Reviewer 1 Report

Comments and Suggestions for Authors

Diffuse large B-cell lymphoma (DLBCL) is the most common lymphoma subtype. Soluble CD163 (sCD163) is shed by M2 tumor-associated macrophages and may have prognostic value in cancers. This study evaluated sCD163 levels in DLBCL patients treated with immunochemotherapy.

In order to improve the quality of the article, the following are suggested:

1.     Introduction:

1. The introduction is not concise and focused enough. Too much background information is provided that could be removed. It is suggested the introduction is made briefer and more focused.

2. The importance and rationale for conducting this study is not clearly stated. It is suggested to better explain in the introduction why this study is important.

3. The hypothesis and objectives of the study are not explicitly stated. It is suggested to clearly state the hypothesis and aims of the study.

4. The structure of the introduction is not cohesive and does not follow a logical flow. It is suggested the structure of the introduction is improved.

Overall, it is suggested the introduction is made more concise, focused, and structured, with greater emphasis on the importance, hypothesis and objectives of the study.

2.     Material and Methods:

Here are the suggestions to improve the quality of the Material and Methods and the overall manuscript:

1. The sampling method is not clearly described. The inclusion and exclusion criteria for patient selection are not specified.

2. Insufficient details are provided on the sCD163 measurement method. No information is given regarding the assay sensitivity and specificity.

3. The statistical methods are briefly explained. More details are needed.

4. Ethical considerations of the study are not mentioned.

Overall, providing more complete details on the methods used can enhance the credibility of the study findings. Clearly outlining the patient selection, assay procedures, statistical analysis, and ethical approval will improve the quality of the manuscript and transparency of the methods.

3.     Material and Methods:

Here are some suggestions to improve the quality of the results section:

1. The results are not presented in a logical flow. Restructuring the order of the results, starting with descriptive statistics, would enhance clarity.

2. Key data is missing from the results, including a table of baseline patient characteristics. This table would help readers interpret the data.

3. The survival analysis methods and outcomes are not described in sufficient detail. More information is needed on the statistical approaches used and the hazard ratios/p-values from survival models.

4. The number of patients included in each analysis is unclear. Patient numbers should be reported for each major analysis.

5. Data visualization is limited. Adding graphs such as Kaplan-Meier curves would help illustrate key findings.

6. Subgroup analyses based on clinical or demographic factors are not presented. These analyses could provide greater insight into the relationship between sCD163 and outcomes.

7. The clinical implications of the findings are not adequately discussed. More interpretation is needed to place the results into context.

In summary, restructuring the order of presentation, adding missing data, expanding details on analyses, improving data visualization, including subgroup findings, and providing more clinical interpretation would enhance the quality of the results section. These changes would help ensure comprehensive reporting of the study findings.

Comments on the Quality of English Language

Here is my assessment of the language quality of the manuscript:

Overall, the language quality is fair but needs improvement in some areas:

- The writing lacks clarity and concision in parts of the text. Some sentences are long and convoluted. The writing could be tightened up and made more precise.

- Grammar, punctuation, and spelling errors are present throughout. Additional proofreading is required to correct these issues. 

-  Simplifying the language would improve readability.

- The structure and logical flow needs refining in some sections like the abstract and introduction. Improved organization is needed. 

To raise the language quality, I recommend a thorough edit focusing on improving clarity, tone/style, grammar, word choice, sentence structure, and flow. Standardizing the formatting and proofreading closely to fix errors will also help elevate the professionalism of the language and aid readability. Implementing these changes will enhance the manuscript considerably.

Reviewer 2 Report

Comments and Suggestions for Authors

The Authors present a clearly analyze of serum soluble CD163 levels in patients with diffuse large B-cell lymphoma treated with chemoimmunotherapy. They suggest that serum soluble CD163 may serve as a novel prognostic indicator in patients with diffuse large B-cell lymphoma. Patints with lower levels of soluble CD163 may have a better prognosis, whereas those with higher levels may be more likely to experience resistance or recurrence. I think that the analyzed group is too small to draw clear conclusions. However, the manuscript is interesting and is important for further research. 

Comments on the Quality of English Language

The manuscript should be edited for proper English language, grammar, punctuation, spelling, and overall style.
